# BENCHMARKING GENERALIZATION OF FOUNDATION MODELS FOR REMOTE SENSING

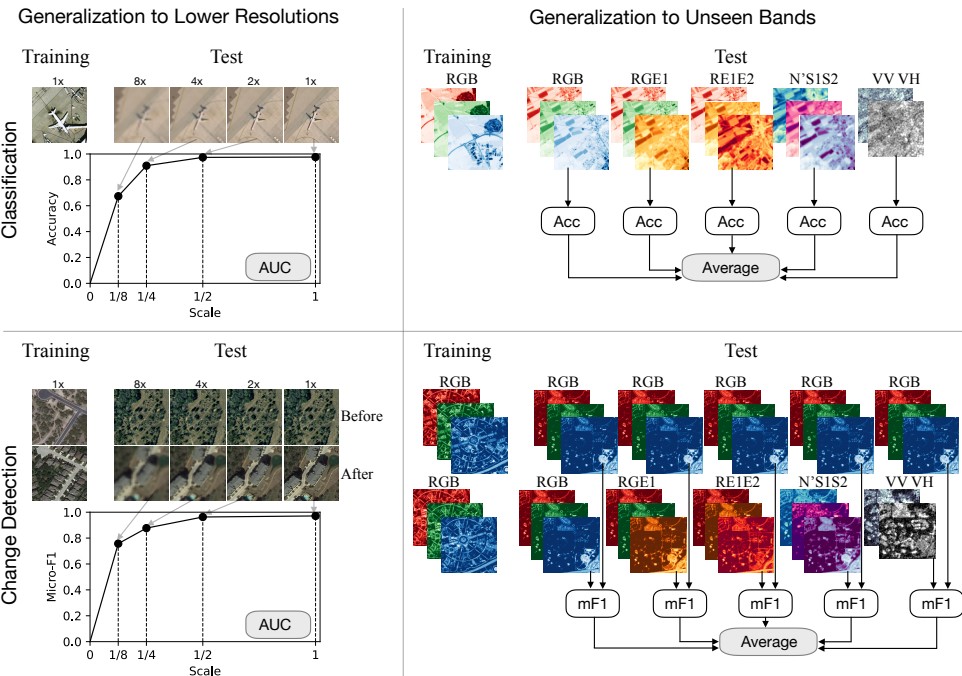

## ABSTRACT

Foundation models have significantly advanced machine learning applications across various modalities, including images. Recently numerous attempts have been made on developing foundation models specifically tailored for remote sensing applications, predominantly through masked image modeling techniques. This work explores the essential characteristics and performance expectations for a foundation model in aerial imagery. We introduce a benchmark designed to evaluate the model's performance as well as robustness to changes in scale and spectral bands of the input. Our benchmarks encompass tasks unique to aerial imagery, such as change detection and scene classification, and utilize publicly available datasets RESISC45, BigEarthNet, LEVIR-CD and OSCD. We evaluate recently proposed foundation models on the benchmark. Furthermore, we explore the impact of various design choices in pretraining and fine-tuning on the performance of the models on our benchmark. Specifically, we pretrain several variations of a self-distillation based self-supervised model on aerial imagery datasets, including one without scale-augmentations and another one with a pretrained mask decoder module.

## 1 INTRODUCTION

The rapid advancements in remote sensing technologies have led to an increased reliance on foundation models for interpreting vast amounts of imagery data captured by satellites (e.g., Sentinel-1, Sentinel-2) (Akiva et al., 2022; Mall et al., 2023; Mañas et al., 2021; Wanyan et al., 2023; Cong et al.,

2022; Reed et al., 2023; Sun et al., 2023; Hong et al., 2024; Muhtar et al., 2023; Mendieta et al., 2023; Tang et al., 2023; Fuller et al., 2023; Bao et al., 2023; Guo et al., 2023; Wang et al., 2023c; Bastani et al., 2023). Usually, this data is raw and unlabeled, whereas creating labels is time-consuming and expensive. For many critical downstream tasks, including change detection, image classification, and semantic segmentation (utilized for tasks such as land cover mapping, flood or disaster monitoring, urban growth analysis, vegetation health monitoring, and terrain analysis), having a large amount of labeled data is crucial to train effective models. In line with recent advancements in self-supervised and semi-supervised learning for vision tasks, the current trend is to train a self-supervised model (either contrastive or based on masked image modeling) which later serves as a backbone for subsequent downstream tasks. Subsequently, a small amount of labeled data can be used to fine-tune this self-supervised learning-based backbone, resulting in a competitive model for specific downstream tasks.

In this work, we focus on evaluating the performance of established foundation models, specifically designed for remote sensing imagery, in the context of scene classification (Cheng et al., 2017; Yang & Newsam, 2010; Sumbul et al., 2019) and change detection (Chen & Shi, 2020; Lebedev et al., 2018; Caye Daudt et al., 2018), by focusing on their generalization capabilities across image resolutions and bands. To analyze the impact of the design choices made in those foundation models, we develop another model using a self-distillation approach.

Our contributions are as follows. (a) We develop a benchmark for remote sensing (RS) foundation models that evaluate them with respect to generalization capabilities of the derived models across scale and input bands. (b) We pretrain several versions of iBOT (Zhou et al., 2022), a self-distillation based ViT (Dosovitskiy et al., 2021), on MillionAID (Long et al., 2021), an aerial imagery dataset, to analyze the impact of design choices on our benchmark. One of the versions includes a pretrained UperNet-like (Xiao et al., 2018) head for segmentation and change detection downstream tasks. (c) We show that the publicly available foundation models we have tested (Bao et al., 2023; Bastani et al., 2023; Mendieta et al., 2023; Zhou et al., 2022) have a lot of room for improvement in obtaining generalization capabilities and transferring those capabilities to downstream models.

## 1.1 RELATED WORK

Some recent developments in the field include various approaches using either supervised or self-supervised learning algorithms. Surprisingly, for some transformer-based models, performance on ImageNet (Deng et al., 2009) in certain instances outperforms those pre-trained on remote sensing imagery (Vanyan et al., 2023a). The effect of pre-training on ImageNet vs a large remote sensing scene recognition dataset is studied in Remote Sensing Pretraining (RSP) (Wang et al., 2023a). To serve as a pre-training dataset, some existing techniques involve gathering data from available open-source large remote sensing datasets and employing it to train the self-supervised algorithm. The two main methods to train self-supervised foundation models are contrastive learning-based methods and generative-based methods (masked image modeling).

The contrastive learning-based approaches include: SECO (Mañas et al., 2021) employs a generalization of contrastive learning, defining various types of augmentations (seasonal, artificial, and mixed). They also created a dataset from Sentinel-2, by first picking locations closer to urban areas and later generating several images for these locations for various seasons. CACo (Mall et al., 2023) introduces a new contrastive loss called Change Aware Contrastive Loss which considers long-term temporal information in satellite imagery to encourage invariance to seasonal variations while maintaining sensitivity to permanent changes. MATTER (Akiva et al., 2022) presents a material and texture-based approach for self-supervised pretraining to generate good representations for remote sensing downstream tasks. Dino-MC (Wanyan et al., 2023) utilizes the DINO pre-training framework for self-supervised learning by using multiple crops of the same image with different sizes. Finally, (Tolan et al., 2024) pretrained DINOv2 on MAXAR imagery with a 0.59m Ground Sample Distance (GSD) and collected a new high-resolution dataset to further enhance performance.

Another stream of works, SatMAE (Cong et al., 2022) Scale-MAE (Reed et al., 2023), RingMO (Sun et al., 2023), SpectralGPT (Hong et al., 2024) extend on Masked Autoencoders (MAE) (He et al., 2022), which is a successful foundation model, based on masked image modeling, where the pretext task is to recover the image, based on its masked version. Scale-MAE (Reed et al., 2023) makes two significant contributions to the MAE (He et al., 2022) framework. First, it introduces the GSD-based

positional encoding. Second, it introduces the Laplacian-pyramid decoder to the MAE framework, encouraging the network to learn multiscale representations. SatMAE (Cong et al., 2022) utilizes temporal and spectral metadata in a positional encoding to encode spatiotemporal relationships in data. RingMo (Sun et al., 2023) modified the masking strategy of MAE (He et al., 2022) to adapt to dense and small objects in complex RS images and trained a self-supervised representation learning model on a dataset of two million unlabeled RS images. SpectralGPT (Hong et al., 2024) is an MAE-based RS foundation model that utilizes a 3D masking for processing spectral data, an encoder to learn spectrally visual representations, and a decoder for multi-target reconstruction. SpectralGPT is pretrained on 1 million multispectral images from the Sentinel-2 satellite (containing 12 spectral bands) from fMoW (Christie et al., 2018) and BigEarthNet (Sumbul et al., 2019).

A more recent direction of works aimed to combine reconstruction-based and contrastive learning-based approaches: CMID (Muhtar et al., 2023) learns representations with both global semantic separability and local spatial perceptibility by combining contrastive learning with masked image modeling in a self-distillation way. GFM (Mendieta et al., 2023) also utilizes a masked image modeling framework. They first gather a pretraining dataset of 1.3 million Sentinel-2 images using the sampling technique from SECO (Mañas et al., 2021). They utilize the dataset to pre-train a Swin-B Transformer model with the MIM objective from SimMIM (Xie et al., 2022). GFM (Mendieta et al., 2023) observed that some of the state-of-the-art methods for aerial imagery often do not perform better than ImageNet-22k pretrained ViTs. Cross-Scale-MAE (Tang et al., 2023) enhances MAE framework by incorporating several additions, including scale augmentations and enforcing cross-scale information consistency to improve its performance across different scales. CROMA (Fuller et al., 2023) encodes masked-out multispectral optical and SAR samples, aligned in space and time to perform cross-modal contrastive learning.

Some other works aim to utilize various channels simultaneously, considering that many satellites can capture multiple bands simultaneously. For example, Sentinel-2 is capable of capturing images with 13 different bands. ChannelViT (Bao et al., 2023) constructs patch tokens independently for each input channel and then uses learnable channel embeddings added to the patch tokens, similar to positional embeddings. ChannelViT generalizes well even when there is limited access to all channels during training. SkySense (Guo et al., 2023) is pretrained on a curated multi-modal RS dataset of 21.5 million multimodal RS image triplets involving RGB high-resolution images and multitemporal, multispectral and SAR sequences. It incorporates a factorized multi-modal spatiotemporal encoder taking temporal sequences of optical and SAR data as input. CROMA (Fuller et al., 2023) and DeCur (Wang et al., 2023c) explored multi-modal pre-training.

Another direction of works focused on multi-task pertaining: Satlas (Bastani et al., 2023) introduced a large dataset for RS supervised pertaining as well as proposed a multi-task model, facilitating training on the multitask annotated Satlas dataset. A more recent, Multi-Task Pretraining (MTP) (Wang et al., 2024) proposes to use a shared encoder and a task specific decoder for pertaining stages to address the issue of transferring the pretrained model into specific downstream tasks. MTP uses the the SAMRS dataset (Wang et al., 2023b), for pre-trained. The SAMRS dataset leverages the segment anything model (SAM) and the well known RS datasets to develop an efficient pipeline for generating a large scale RS segmentation dataset.

Recently, for the change detection task, an end-to-end super-resolution-based network for high-resolution image change detection SRCDNet (Liu et al., 2022), was developed to address the change detection problem, for various resolution images. We extend this idea to more classification and change detection datasets.

## 2 EXPECTATIONS FROM FOUNDATION MODELS

Most of the evaluation strategies for remote sensing foundation models are based on fine-tuning on downstream classification, segmenetation or other datasets, and measuring the performance on the corresponding test sets. While this is an important aspect, it does not capture the potential benefits of large-scale pretraining.

We argue that the foundation models should bring features and capabilities to the fine-tuned models that would not be possible by leveraging solely the labeled data. These capabilities include general-

ization across various kinds of shifts. In this paper we focus on two axis of generalization: image resolution (i.e. ground sampling distance) and the set of bands of the capturing device.

A notable example from natural language processing is described in (Garcia et al., 2023). A large language model is trained on a mixture of non-parallel English and Chinese texts, and then it is prompted to translate with only five pairs of translated sentences. With such prompts the language model almost matches the performance of Google Translate production model. Hence, its translation capabilities generalize to vocabularies and topics way beyond the five examples can cover.

There are many axes of variation in aerial imagery: resolution, weather conditions, time of the year, time of the day, geographical location etc. We expect strong foundation models for remote sensing to enable models derived from them to generalize across all possible variations. New benchmarks are required to measure this kind of generalization.

Note that in natural language processing, large language models pretrained on web-scale data, allow few-shot learning, where the downstream task is described by a few samples written in the prompt, or even zero-shot learning, where the task is described in human languages without explicit examples. This aspect is not measured for pure image-based methods as they do not have an interface for describing the task at the input. Vision-language models like CLIP (Radford et al., 2021) or Chameleon (Team, 2024) allow evaluations of similar capabilities, but their analysis is beyond the scope of this work.

This implies that pure image-based models require fine-tuning of some form to be adapted for a downstream task. We are interested in measuring the generalization of the adapted models. Hence, the way the models are adapted can be critical in retaining generalization capabilities. Developers of the foundation models should ideally provide recipes for adaptation on downstream tasks that preserve the generalization properties.

**Compute constraints.** Foundation models should target specific compute requirements. Many downstream applications require the models to run on low power devices or need to support large volumes of data in deployment, and hence require limited number of FLOPs per image. It is important to note that this requirement refers to the fine-tuning stage and the deployment of the final model, and not to the pretraining process. For example, DINOv2 (Oquab et al., 2023) has a ViT-B version which is distilled from a larger ViT-g model. While the large model was trained using hundreds of GPUs, the distilled version can be easily fine-tuned on a single consumer-grade GPU.

In summary, a typical foundation model for remote sensing can use unlimited compute and data for pretraining, but should be able to run on limited compute during inference, should support a simple recipe for fine-tuning for any downstream task, and, most importantly, should inherit remote-sensing specific generalization capabilities. We attempt to formalize these expectations by designing a benchmark in the next section.

## 3  BENCHMARK DESCRIPTION

In this section, we propose a benchmark suite for evaluating foundation models for remote sensing in the spirit of the expectations defined in Section 2.

### 3.1  CHOOSING AXES OF GENERALIZATION

In this subsection we analyze possible axes of generalization for foundation models.

**Generalization to lower spatial resolutions.** This is an important direction with many practical applications. Low resolution satellites like Landsat and Sentinel constantly provide publicly available imagery, while higher resolution imagery is usually harder to find. In many scenarios image labeling is being done on higher quality imagery, but in test time the images might come from satellites with lower resolution. We expect the models to perform on such distribution shifts as good as on the images with the original resolution.

**Generalization to higher spatial resolutions.** This scenario might also happen in practical applications, but retaining the original performance on higher resolution is trivial by downsizing the images to the original resolution. One can expect that the additional details visible in higher resolution

might allow to *exceed* the performance on the original images, but this is beyond the scope of a generalization benchmark.

**Generalization to other bands.** This is very practical and is covered by our proposed benchmark. In remote sensing, different satellites capture imagery across various spectral bands. Often, models are trained on imagery from a limited number of bands (e.g., RGB or a specific multispectral range), but in real-world applications, they may encounter images with additional or fewer bands. The ability to generalize across different spectral compositions is crucial as additional bands may provide complementary information as well as for some applications some bands may be missing.

**Generalization to other seasons.** In important applications, such as change detection or segmentation, a good foundation model should generalize well across seasons. For example, when identifying areas with new construction, whether the landscape is snowy, sunlit or foggy should not affect the model's performance. This is important, but is challenging to collect data because the seasons are not coherent in different geographical regions. This is left for future work.

**Generalization to other times of the day.** Similar to season generalization, a good foundation model should generalize well across different times of the day. Variations such as changes in shadows, lighting conditions, and overall brightness should not impact the model's performance. However, this is not a practical problem, as even the same satellite will visit the same location at different times of the day, so every practical dataset will have internal variability across this dimension.

**Generalization to other geographical locations.** A strong foundation model should be able to generalize accross different geographical regions, as terrain types, nature and human-made structure/architecture change significantly based on location. Evaluating this kind of generalization is notoriously hard because of significant label shift that exists between geographic locations which strongly affects the performance of the models. FMoW-WILDS (Koh et al., 2021) dataset contains geographical split, but due to severe shifts in label distributions it is hard to isolate and properly measure the generalization abilities with respect to pure domain shift.

Hence, in this work we focus on two shifts: generalization to unseen resolutions and bands. We will measure generalization on two types of tasks: scene classification and change detection.

## 3.2 Generalization to Smaller Image Resolution

### 3.2.1 Scene Classification

We take two commonly used benchmark datasets in the literature: RESISC45 (Cheng et al., 2017) and UC Merced (Yang & Newsam, 2010) see Appendix A.

We measure the performance not only on the original resolutions of the images, but also on the images with 1/2, 1/4 and 1/8 resolutions. The images are resized by 1/x factor and the scaled back by x which produces an image with the same number of pixels but with lower quality. This mimics how the image could have been captured if the satellite had a lower resolution. As an evaluation metric, we draw a curve where x-axis is the scaling parameter $(1/8, 1/4, 1/2, 1)$ and y-axis is the accuracy score for each version. We report the area under this curve as our final metric, and call it **AUC-Acc**.

In this benchmark we restrict the models to use 50 GFLOPs on a single image. This threshold is independent from the neural architecture, and ViT-B/16 on an image of size 256x256px is within the limits.

### 3.2.2 Change Detection

For change detection we use another two commonly used datasets: CDD (Lebedev et al., 2018) and LEVIR-CD (Chen & Shi, 2020) see Appendix A.

We create partially scaled versions of the test sets of these datasets. We maintain the scale of the first image unchanged, while for the second image, we distort it by reducing its quality by a factor of 2, 4, and 8. Note that a similar setup has been first proposed in (Liu et al., 2022). We evaluate on the original resolution, as well as on the scaled versions. We compute micro-averaged F1 score for each of the versions. Finally we draw a curve where x-axis is the scaling parameter and y-axis is the micro-averaged F1 score for each version. We report the area under this curve as our final metric, and call it **AUC-F1**.

For this benchmark, we restrict the models to use 100 GFLOPs on a pair of images.

### 3.3 Generalization to Unseen Bands

Most of the publicly available multi-band remote sensing datasets originate from European Sentinel satellites. Sentinel-1 uses C-band synthetic aperture radar (SAR) and captures VH and VV bands that store complex numbers. Different papers use various preprocessing schemes for those values. In order to be able to merge images from various sources, we use the absolute values of the complex numbers, and do not perform any additional preprocessing. Hence, for SAR data we have two bands, denoted by VV and VH. Sentinel-2 has 12 bands of varying resolution. Following (Bao et al., 2023) and many other papers, we drop 60m resolution bands (e.g. "coastal"), and use the bands with 10m and 20m resolution (resizing all of them to 10m). Details are in Table 4 in Appendix.

#### 3.3.1 Scene Classification

To create a benchmark for evaluating the generalization on unseen bands for a classification task, using the BigEarthNet (Sumbul et al., 2019) dataset, we utilize BigEarthNet-medium, which contains approximately 10% of the images from the original BigEarthNet dataset. This dataset is created for multi-label classification. First, we remove the images tagged as clouds and snow using the lists available at http://bigearth.net. We use the BigEarthNet-medium dataset as follows: for each experiment, we train on the RGB channels and evaluate on four tri-channel triplets and one bi-channel pair: RGB, RGE1, RE1E2, N'S1S2, and VV VH (bi-channel). We compute the micro average precision (mAP) for each experiment and report the average over these five values. The goal is to determine if a model trained on RGB channels can generalize to other channel combinations.

#### 3.3.2 Change Detection

To create a benchmark for evaluating the generalization on unseen bands for a change detection task, we use the Onera Satellite Change Detection (OSCD) dataset see Appendix A.

## 4 Factors Contributing to the Performance

### 4.1 iBOT pretraining

To perform analysis of various factors on the generalization capabilities of the fine-tuned models, we pre-trained several iBOT models using satellite imagery. As demonstrated in (Vanyan et al., 2023a), self-distillation-based models, such as iBOT, outperform MIM-based models in obtaining robust image representations, even in satellite imagery. Drawing on the findings of (Vanyan et al., 2023a), we selected iBOT for pre-training with the MillionAID dataset (Long et al., 2021). To accommodate the varying image sizes, we divided the original images into smaller square tiles, with each side limited to a maximum of 550 pixels, resulting in a total of 2106700 images. Note that even the original iBOT pretrained on ImageNet is quite strong, so we also included it in our comparative analyses.

We trained iBOT for 200 epochs with peak learning rate $5 \times 10^{-4}$ that linearly decreases to $2 \times 10^{-6}$ over 5 warmup epochs. All RandomResizeCrops were converted to RandomCrops in the transforms. The training was conducted using PyTorch Distributed Data Parallel to utilize multiple GPUs and used 100 batch size per GPU. The experiments were performed on NVIDIA DGX A100 at the local university and an instance with 8 NVIDIA H100s kindly provided by Nebius.ai. The loss curve followed the typical pattern of similar networks (Fig. 4 in Appendix). The resulting model is labeled as iBOT-MillionAID.

### 4.2 Augmentation

Here we analyze the impact of scale augmentation on the robustness to scale changes. The original iBOT algorithm has an augmentation module that randomly resizes the pictures and then crops to a fixed size. For this experiment we have pretrained two versions, one without resizing, i.e. the scale augmentation, and another one with resizing. The hypothesis is that the pretrained model will be

more robust to scale changes, and this robustness will be transfered to the fine-tuned models, which will cause higher AUC scores on our benchmarks.

Furthermore, we perform experiments with scale augmentation during fine-tuning. In this setting, we randomly shrink the image (or the second image in case of change detection) by 2, 4 and 8 times, and then resize it to get to the original resolution. This is the same transformation as we did when transforming the test sets for our benchmark. While this kind of augmentations during fine-tuning are beyond the scope of our benchmarks, the results of these experiments can act as an upper bound for the scale robustness of the models.

Table 7 in Appendix shows the full results. When augmentations are not applied during fine-tuning, augmentations during pretraining at 1:1 and 1:2 resolutions consistently give better results across all datasets. However, this trend does not hold for smaller resolutions.

On the other hand, augmentations during fine-tuning have a significantly higher impact on the generalization. In case of classification, we leverage $2\times$, $4\times$, and $8\times$ versions of the original dataset. Although we obtain $4\times$ more data, this does not add new information, and we keep the total number of optimization steps constant by decreasing the number of epochs by $4\times$. In case of change detection, we randomly choose one of the augmented versions of the second image at each epoch, and train for the same number of epochs as in the experiment without augmentations.

These experiments indicate that scale augmentation during pretraining still does not produce generalization capabilities at a level comparable to what one can obtain by augmenting during fine-tuning.

## 4.3 Pretrained Mask Decoder

Many downstream tasks in the remote sensing domain require an additional module on top of the backbone to produce a binary mask. These include segmentation tasks that work on a single input image and change detection tasks that require two input images. Here we develop an extension of iBOT-MillionAID to have an additional mask decoder module that is already pretrained on large amounts of data. As MillionAID does not contain any segmentation or change masks, we leverage the teacher-student structure of iBOT and artificially generate masks the following way. The original iBOT implementation passes two *global crops* of the input image to the teacher and the student, and additional ten *local crops* to the student. We draw the mask of the second global crop in the coordinate space of the first global crop and store it as a target mask. The patch representations of the first global crop from the teacher and the second global crop by the student are concatenated and passed to an UperNet decoder (Xiao et al., 2018) which produces a binary mask. This module adds an additional pixel-wise cross-entropy loss term. Note that UperNet's inputs come from four ViT layers (3rd, 5th, 8th and 12th), not only the last one.

We explored the joint training of UperNet and the regular iBOT. We investigated two methods to integrate mask loss into the iBOT training: either by obtaining the patch representations of both global crops using only the student model or by using the teacher model to obtain one of them. The training loss of the first approach was unstable, with some of the layer activations increasing significantly during the training. The teacher-student approach didn't encounter these issues, resulting in successful joint training. The final architecture is shown in Fig. 1.

Table 1: The effect of a pretrained mask decoder on change detection tasks. All models are iBOTs pretrained on MillionAID with scale augmentation.

| LEVIR-CD | 1:1 | 1:2 | 1:4 | 1:8 | AUC-F1 |
|---|---|---|---|---|---|
| Without Mask Decoder | $90.6 \pm 0.2$ | $87.6 \pm 0.9$ | $50.4 \pm 15.1$ | $2.0 \pm 1.0$ | $65.2 \pm 3.2$ |
| With Mask Decoder | $90.6 \pm 0.1$ | $89.2 \pm 0.1$ | $66.6 \pm 5.0$ | $4.3 \pm 1.1$ | $69.1 \pm 1.0$ |
| **CDD** | | | | | |
| Without Mask Decoder | $97.4 \pm 0.0$ | $96.8 \pm 0.0$ | $91.4 \pm 0.6$ | $79.2 \pm 0.9$ | $87.7 \pm 0.2$ |
| With Mask Decoder | $97.1 \pm 0.0$ | $96.7 \pm 0.0$ | $91.5 \pm 0.5$ | $80.1 \pm 0.9$ | $87.7 \pm 0.2$ |

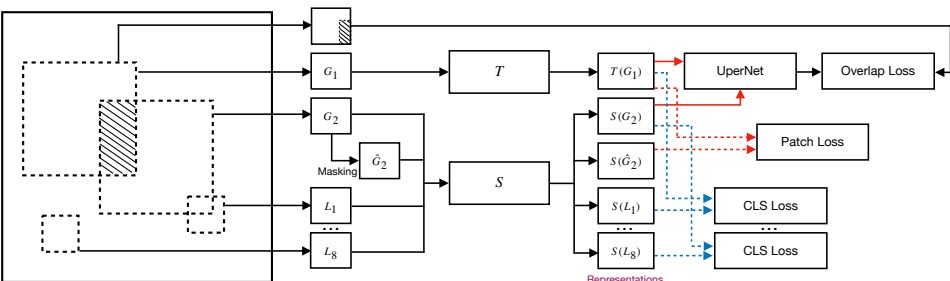

Figure 1: iBOT pretraining architecture with an additional UperNet mask decoder that is trained using the "overlap loss". There are two global and eight local crops of the original image that pass through Teacher (T) and Student (S) networks. The loss terms are calculated on top of various parts of the extracted representations. Dotted lines imply that only the representations of the last layers are used. Solid lines imply that representations of four layers are used (as an input to UperNet). Red lines correspond to patch representations, while the blue lines correspond to CLS vectors.

We used $2.5 \times 10^{-4}$ peak learning rate and cosine decay with 5 warmup epochs. We trained the model for approximately 800 H100 GPU hours on an instance with 8 NVIDIA H100s provided by Nebius.ai.

As shown in Table 1, there is a slight improvement in performance and significantly lower variance across all scales with the pretrained mask decoder on LEVIR-CD. There is no visible change on CDD. This can be explained by the large size of the CDD dataset. Similar to the discussion in Section 2, it is likely that the additional power of the pretrained models is not critical when the fine-tuning dataset is large enough. Another potential way to enhance the impact of pretrained decoders is to pretrain it with denser supervision signal. While we used a binary mask calculated during pretraining, (Wang et al., 2024) uses segmentation pseudo-labels generated by a strong domain-agnostic segmentation model. The impact of that kind of supervision signal during pretraining is left for future work.

### 4.4 CATASTROPHIC FORGETTING DURING FINE-TUNING

While the pretrained models can have inherent generalization capabilities, it is possible that the models "forget" those during fine-tuning. One way to measure this phenomenon is to repeat the fine-tuning experiments with frozen backbones. In this setting the only part of the model that has never seen inputs of diverse scales is the final linear layer (in case of classification) or the decoder.

Table 2 shows that the effect strongly depends on the downstream dataset. Particularly for RESISC45, the approach with frozen backbone is significantly more robust to lower resolutions than the one with full fine-tuning. On LEVIR-CD the same trend can be observed on 1:4 and 1:8 resolutions, but the performance of the model with frozen backbone is slightly worse on 1:1 and 1:2 resolutions compared to full fine-tuning. On UC Merced we see the oposite behaviour, when freezing the backbone enhances performance on higher resolutions, but on lower resolutions full fine-tuning outperforms the frozen model.

## 5 BASELINES

For the benchmarks on generalization to lower resolutions, we used SatlasPretrain (Bastani et al., 2023) trained on high-resolution imagery (Aerial) and on the RGB subset of Sentinel-2 imagery (Sentinel2), GFM (Mendieta et al., 2023), and general-purpose iBOT pretrained on ImageNet as baseline. For the benchmarks on generalization to unseen bands, we used ChannelViT (Bao et al., 2023) and SatlasPretrain's multispectral version. Each of these models has a different training paradigm and pretraining dataset. iBot is a self-supervised method pretrained on ImageNet. GFM combines two concepts: self-supervised pretraining on a custom-collected dataset, GeoPile, and continual pretraining to retain knowledge obtained from pretraining on ImageNet. SatlasPretrain is pretrained on a custom-collected dataset, Satlas, in a supervised manner. ChannelViT is a supervised method that considers the presence of a varying number of bands in the input data. Clay v1 (cla,

Table 2: The impact of full fine-tuning on the loss of generalization capabilities. All models are iBOTs pretrained on MillionAID with scale augmentation. No scale-augmentation was performed during fine-tuning (or linear probing).

| **RESISC45** | | | | | AUC-ACC |
|---|---|---|---|---|---|
| Full fine-tuning | $93.4 \pm 0.2$ | $84.3 \pm 1.2$ | $47.4 \pm 5.6$ | $18.7 \pm 2.0$ | $66.2 \pm 1.8$ |
| Frozen backbone | $\mathbf{94.6 \pm 0.1}$ | $\mathbf{92.2 \pm 0.2}$ | $\mathbf{66.5 \pm 1.5}$ | $\mathbf{25.1 \pm 1.3}$ | $\mathbf{73.8 \pm 0.5}$ |
| **LEVIR-CD** | 1:1 | 1:2 | 1:4 | 1:8 | AUC-F1 |
| Full fine-tuning | $90.6 \pm 0.2$ | $87.6 \pm 0.9$ | $50.4 \pm 15.1$ | $2.0 \pm 1.0$ | $65.2 \pm 3.2$ |
| Frozen backbone | $84.4 \pm 0.0$ | $84.4 \pm 0.2$ | $61.6 \pm 7.8$ | $3.4 \pm 4.0$ | $64.7 \pm 2.0$ |
| **UC Merced** | | | | | AUC-ACC |
| Full fine-tuning | $98.7 \pm 0.8$ | $97.9 \pm 1.3$ | $84.3 \pm 4.3$ | $46.0 \pm 8.3$ | $82.9 \pm 1.0$ |
| Frozen backbone | $99.5 \pm 0.1$ | $99.2 \pm 0.3$ | $75.7 \pm 2.9$ | $31.3 \pm 3.9$ | $80.2 \pm 0.7$ |

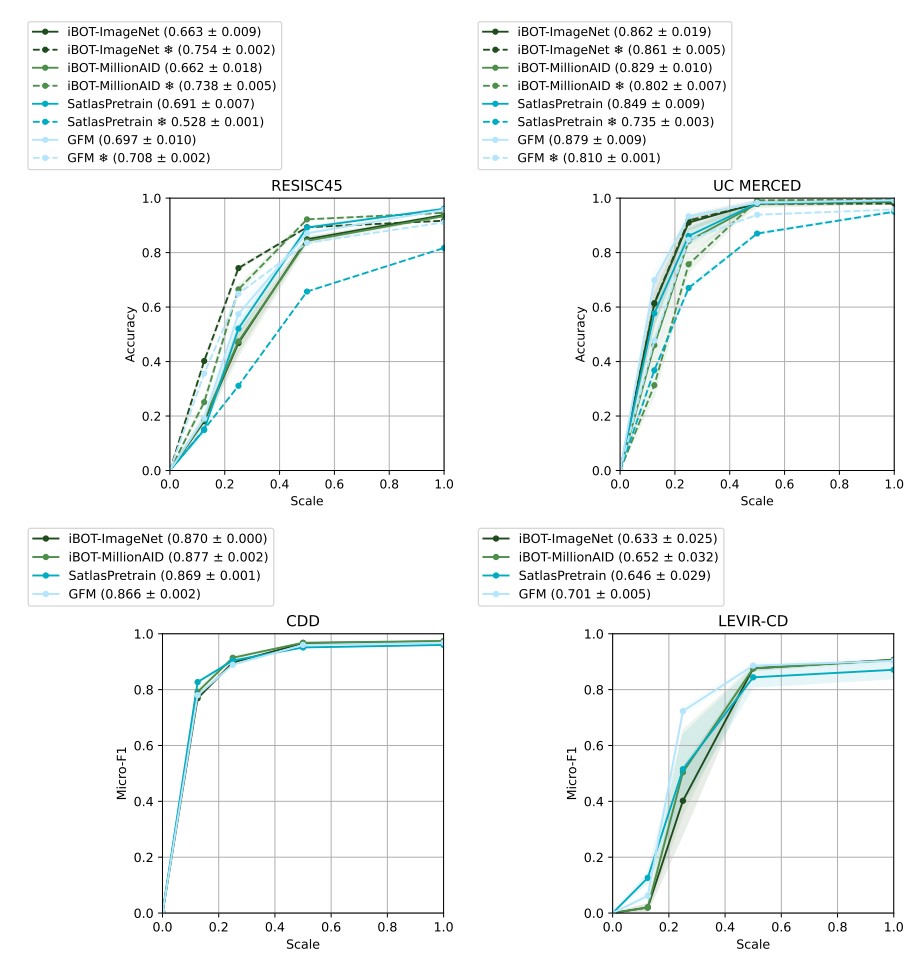

Figure 2: The results of the baselines on our benchmark tasks for generalization across image resolution. The top row shows classification on RESISC and UC Merced, while the bottom row shows change detection on CDD and LEVIR-CD. X-axis: Scale of Distortions, Y-axis: Micro-F1 Scores.

2024) is a self-supervised method that utilizes a hybrid loss combining distillation and reconstruction components. This model also accepts a varying number of input channels. Prithvi (Jakubik et al., 2023) is a modification of a MAE model to support 3D inputs with 6 channels.

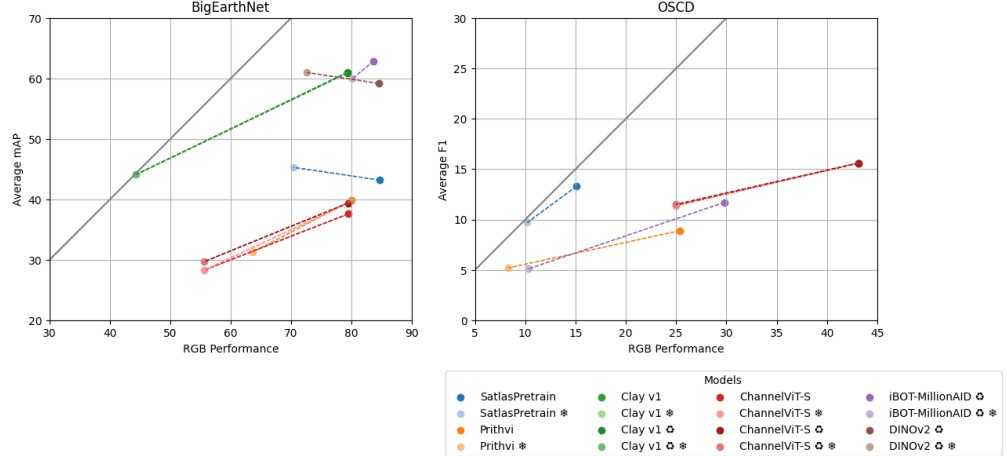

Figure 3: The results of the baselines on our benchmark tasks for generalization across bands. X-axis shows the performance on the RGB bands (the ones used in fine-tuning), while Y-axis shows the average performance as defined in the benchmark. For more details see Table 3 in Appendix.

## 5.1 Experimental Setup

To adapt the models for classification, we add a linear layer on top of the [CLS] token representation, if available, or on top of the global average pooled vector of all patch representations.

To test the models for change detection, we take the backbone, which is either a Swin Transformer, or a ViT, and integrate the UperNet head (Xiao et al., 2018). The two source images go through identical backbones, and the resulting representations are substracted from each other and passed to the head. In the case of ViTs, we use an additional *neck* module between the backbone and UperNet. The backbone is initialized with the pre-trained weights and further fine-tuned using the change detection datasets. In case of our iBOT trained on MillionAID, the neck and the head modules are also initialized, and we take the concatenation of features instead of the difference. In the experiments involving the ChannelViT backbone we sum up representations along the channel axis before passing it to UperNet decoder. For more details see Appendix B.

## 5.2 Results and Conclusions

The results are shown in Figures 2 and 3, and Tables 3 and 5 in Appendix. The general conclusion is that all tested models struggle with generalizability both across scales and bands.

There are cases where the same model with a frozen backbone performs slightly better than its fine-tuned counterpart (e.g. SatlasPretrain and DINOv2). For generalization across bands, the fine-tuned models are always better for the RGB, but may fall behind frozen models on unseen bands. The performance gap between frozen models and full fine-tuning is relatively large for ChannelViT-S, Prithvi and especially Clay v1. This performance drop is also noticeable in classification tasks during scale evaluation with the SatlasPretrain model, which could be due to its supervised pretraining. However, we can observe that, when fine-tuning on larger datasets, the weakness of supervised pretraining becomes less significant, as seen with SatlasPretrain on BigEarthNet.

ChannelViT performs quite poor on BigEarthNet, which can be explained by the relatively small size of the model. We hypothesize that models pretrained in a self-supervised manner require less data for the downstream tasks compared to those pretrained in a supervised manner. Since Prithvi is an extension of the MAE model, this may explain why its performance drops during linear probing. As mentioned in some studies (He et al., 2022; Vanyan et al., 2023b), models trained with masked image modeling exhibit their advantages when fully fine-tuning them; their representations are not designed for linear probing. Finally, we note that the models that were pretrained on multiple bands are not able to leverage the knowledge on the extra bands they learned during pretraining. When the unseen bands are given in the place of RGB bands (♻), the results are not worse compared to the case when the unseen bands are given in their original input locations. This opens a wide avenue for future work.

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

## A DATASETS

RESISC45 (Cheng et al., 2017) and UC Merced (Yang & Newsam, 2010) datasets contain 256x256px images. Image resolution is 30cm/px for UC Merced and varies 20-600cm/px for RESISC45. Both datasets use RGB bands only. We take the splits defined in (Neumann et al., 2019).

The LEVIR-CD dataset (Chen & Shi, 2020) comprises a substantial collection of bitemporal Google Earth images. It includes 637 image pairs, each sized $1024 \times 1024$px, with $400$ images designated for training. The images in the training set have a resolution of 50cm/px. Originating from 20 distinct regions within cities in Texas, USA, these images showcase the construction-induced changes. The fully annotated LEVIR-CD dataset encompasses a total of $31,333$ individual changed buildings. The changes in the LEVIR-CD dataset primarily come from the construction of new buildings. The average size of each changed area is approximately $987$ pixels.

The CDD (Lebedev et al., 2018) dataset contains season-varying remote sensing images of the same region, obtained from Google Earth (DigitalGlobe). The dataset comprises $16,000$ image sets (two images of the same location and the annotated change), each with an image size of $256 \times 256$ pixels and 0.03-1m/px ground sample distance.

Onera Satellite Change Detection (OSCD) dataset contains pairs of aerial images of the same location captured at different times, with changes manually annotated at the pixel level (Caye Daudt et al., 2018). The dataset contains images from a total of 24 cities, divided into smaller chunks ($192 \times 192$) of images. Similar to the classification benchmark, we train on the RGB channels and evaluate on four tri-channel triplets and one bi-channel pair: RGB, RGE1, RE1E2, N'S1S2, and VV VH (bi-channel). We note that for the evaluation, we always keep the first picture as RGB and the second figure with the corresponding band channels. We compute the micro F1 score for each experiment and report the average over these five values.

## B IMPLEMENTATION DETAILS

All the codes for pretraining, as well as the benchmarks proposed by us with all the hyperparameters, can be found at: `https://anonymous.4open.science/r/rs_foundation_models-42DC/README.md`.

### B.1 CLASSIFICATION

We perform two kinds of fine-tuning: full fine-tuning and linear probing. For both setups, we train for 100 epochs. For all experiments in the full fine-tuning setup or linear probing, we evaluate using the last checkpoint. However, for full fine-tuning on the BigEarthNet dataset, we select the best checkpoint based on performance on the validation set. In all experiments within the full fine-tuning setup, we use the $AdamW$ optimizer with a learning rate of $10^{-4}$ employing $WarmupCosineAnnealing$ scheduling and an estimated minimum value of $10^{-5}$. In experiments within the linear probing setup, we use the $AdamW$ optimizer with a learning rate of $10^{-3}$ employing $MultiStep$ scheduling and an estimated minimum value of $10^{-5}$.

In the linear probing setup for the Prithvi model, we conducted a grid search to optimize the hyperparameters. The optimization process involved testing three different optimizers: $\{Adam, AdamW, SGD\}$. For the learning rate, we evaluated three values: $\{10^{-3}, 10^{-4}, 10^{-6}\}$ setting one of the following schedulers: $\{MultiStep, WarmupCosineAnnealing\}$. We selected the $AdamW$ optimizer with a learning rate of $10^{-3}$ and the $WarmupCosineAnnealing$ scheduler for our final configuration based on the performance on the validation set. For linear probing with the ChannelVit model, we use the initial hyperparameters for linear probing provided by the authors and perform the same grid search. Ultimately, we choose the $Adam$ optimizer with an initial learning rate of $10^{-3}$ and $MultiStep$ scheduling.

### B.2 CHANGE DETECTION

For change detection experiments, we train our models for 200 epochs. We use the $AdamW$ optimizer with a learning rate of $6 \times 10^{-5}$ along with $WarmupCosineAnnealing$ which includes warmup steps of 10 and batch size of 32. For experiments on OSCD dataset we choose learning rate $3 \times 10^{-5}$ decrease the training epochs to 100 and use warmup steps of 5 with a batch size of 4.

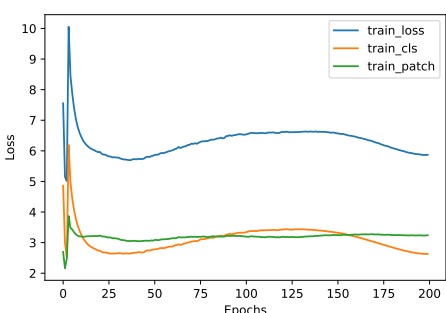 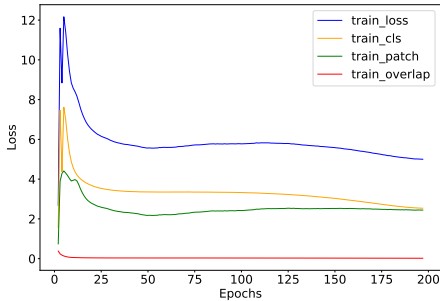

Figure 4: Overall loss and loss components of the iBOT trained on MillionAID dataset for 200 epochs with scale augmentation and without a mask decoder on the left and with mask decoder on the right.

## C  DETAILED RESULTS

In Table 5 we present the benchmark results for proposed and existing models in change detection (LEVIR-CD and CDD) and classification (RESISC45 and UC Merced). For classification, we demonstrate results for both full fine-tuning and linear probing. All experiments are conducted with scale distortions of 1:1, 1:2, 1:4, and 1:8. The AUC-F1 score is reported for change detection, and the AUC-ACC score is reported for classification. For change detection, we compare iBOT trained on ImageNet, our trained iBOT for MillionAID, Satlas, and GFM. For the LEVIR-CD dataset, the results are generally comparable across methods. However, GFM shows a clear advantage over the other methods for the 1:2 and 1:4 scale distortions. Specifically, while all four methods produce comparable results at 1:2, GFM demonstrates a clear advantage at 1:4. However, we remark that the pretraining dataset for GFM GeoPile contains RESISC45, which could possibly cause its superior performance over the other methods. For CDD dataset, we observe that all the results are comparable, however, we observe that GFM does not have superior performance over the other methods. The little AUC-F1 score difference between various scale distortions could be explained by the fact that the CDD dataset contains samples from different GSD (0.03m-1m). For classification, we compare iBOT trained on ImageNet, our trained iBOT for MillionAID, the two versions of Satlas and GFM. We observe that for iBOT (both trained on ImageNET and MillionAID) linear probing has a clear advantage over full-finetuning for lower resolutions.

In Table 6, we report the performance of our trained iBOT on the MillionAID dataset, comparing results with and without augmentations, as well as between a frozen backbone or linear probing and full fine-tuning. For change detection on the LEVIR-CD dataset, we observe that full fine-tuning has a clear advantage over a frozen backbone. Additionally, we note that augmentations do not improve performance for this task. For the classification task (RESISC45 and UC Merced), we observe that for both full fine-tuning and linear probing the model trained with augmentations has a clear advantage over the one trained without augmentation.

Experiments with augmentations and the results of the default setup for RESISC45 and CDD datasets show that the diversity of the dataset in terms of real resolutions (GSD) improves the generalization capabilities of the finetuned model, even if the backbone weights are frozen.

Table 4 lists the band names with descriptions, their corresponding names in Sentinel, and the names used in this paper to avoid confusion.

In Figure 4 the left subfigure shows the iBOT loss (total training loss and its components) trained on the MillionAID dataset. The right subfigure displays the iBOT loss (total training loss and its components: train cls, train patch, and train overlap) for the model trained on the MillionAID dataset with the additional mask decoder proposed by us.

Table 3: Generalization to unseen bands of several baselines on two datasets. ♻ indicates that the input channels are treated as RGB channels.

| Dataset | RGB | RGE1 | RE1E2 | N'S1S2 | VV VH | |
|---|---|---|---|---|---|---|
| **BigEarthNet** | | | | | | **Average mAP** |
| SatlasPretrain | $84.7 \pm 0.1$ | $30.9 \pm 1.8$ | $32.8 \pm 1.1$ | $28.7 \pm 1.7$ | $18.7 \pm 1.1$ | $43.2 \pm 0.7$ |
| SatlasPretrain ✳ | $70.4 \pm 0.0$ | $37.6 \pm 0.1$ | $42.5 \pm 0.2$ | $34.3 \pm 0.2$ | $24.0 \pm 0.1$ | $45.3 \pm 0.1$ |
| Prithvi | $80.0 \pm 0.2$ | $31.7 \pm 2.6$ | $34.0 \pm 3.4$ | $28.1 \pm 1.8$ | $25.4 \pm 3.2$ | $39.8 \pm 0.5$ |
| Prithvi ✳ | $63.7 \pm 0.0$ | $24.4 \pm 0.0$ | $24.9 \pm 0.0$ | $27.4 \pm 0.0$ | $16.2 \pm 0.0$ | $31.3 \pm 0.0$ |
| Clay v1 | $79.3 \pm 0.1$ | $72.2 \pm 0.5$ | $57.1 \pm 1.1$ | $52.2 \pm 1.0$ | $44.3 \pm 2.7$ | $61.0 \pm 0.8$ |
| Clay v1 ✳ | $44.3 \pm 0.2$ | $44.3 \pm 0.2$ | $44.3 \pm 0.2$ | $44.3 \pm 0.2$ | $43.2 \pm 0.2$ | $44.1 \pm 0.2$ |
| Clay v1 ♻ | $79.3 \pm 0.1$ | $72.1 \pm 0.4$ | $57.1 \pm 1.1$ | $51.9 \pm 1.1$ | $44.3 \pm 2.7$ | $60.9 \pm 0.8$ |
| Clay v1 ♻✳ | $44.3 \pm 0.2$ | $44.3 \pm 0.2$ | $44.3 \pm 0.2$ | $44.3 \pm 0.2$ | $43.2 \pm 0.2$ | $44.1 \pm 0.2$ |
| ChannelViT-S | $79.5 \pm 0.4$ | $30.6 \pm 2.6$ | $26.1 \pm 3.3$ | $27.1 \pm 5.4$ | $24.8 \pm 2.5$ | $37.6 \pm 1.1$ |
| ChannelViT-S ✳ | $55.6 \pm 0.0$ | $27.7 \pm 0.0$ | $11.3 \pm 0.1$ | $21.5 \pm 0.2$ | $25.3 \pm 0.2$ | $28.3 \pm 0.1$ |
| ChannelViT-S ♻ | $79.5 \pm 0.4$ | $31.1 \pm 2.7$ | $29.7 \pm 2.3$ | $33.5 \pm 1.8$ | $23.3 \pm 1.9$ | $39.4 \pm 1.4$ |
| ChannelViT-S ♻✳ | $55.6 \pm 0.0$ | $27.6 \pm 0.0$ | $22.7 \pm 0.1$ | $26.9 \pm 0.0$ | $15.9 \pm 0.0$ | $29.7 \pm 0.0$ |
| iBOT-MillionAID ♻ | $83.7 \pm 0.1$ | $62.0 \pm 2.3$ | $65.8 \pm 2.4$ | $45.5 \pm 1.8$ | $22.6 \pm 2.6$ | $62.9 \pm 1.4$ |
| iBOT-MillionAID ♻✳ | $80.2 \pm 0.0$ | $60.8 \pm 0.1$ | $61.6 \pm 0.1$ | $41.1 \pm 0.1$ | $25.6 \pm 0.3$ | $60.0 \pm 0.0$ |
| DINOv2 ♻ | $84.5 \pm 0.1$ | $66.9 \pm 2.1$ | $52.9 \pm 2.2$ | $43.9 \pm 2.1$ | $25.0 \pm 0.7$ | $59.2 \pm 1.3$ |
| DINOv2 ♻✳ | $72.6 \pm 0.0$ | $67.2 \pm 0.0$ | $61.3 \pm 0.1$ | $55.8 \pm 0.1$ | $32.6 \pm 0.1$ | $61.0 \pm 0.0$ |
| **OSCD** | | | | | | **Average F1** |
| Prithvi | $25.4 \pm 3.9$ | $2.7 \pm 1.3$ | $2.0 \pm 1.4$ | $3.7 \pm 2.0$ | $10.9 \pm 2.4$ | $8.9 \pm 1.0$ |
| Prithvi ✳ | $8.3 \pm 1.4$ | $4.0 \pm 2.5$ | $3.6 \pm 2.9$ | $4.9 \pm 3.3$ | $5.0 \pm 3.9$ | $5.2 \pm 1.4$ |
| SatlasPretrain | $15.1 \pm 3.0$ | $14.2 \pm 3.3$ | $14.0 \pm 3.4$ | $11.0 \pm 1.9$ | $9.1 \pm 2.8$ | $13.3 \pm 2.5$ |
| SatlasPretrain ✳ | $10.2 \pm 1.5$ | $10.4 \pm 1.2$ | $10.4 \pm 0.7$ | $9.7 \pm 0.3$ | $6.9 \pm 0.5$ | $9.7 \pm 0.5$ |
| ChannelViT-S | $43.1 \pm 1.4$ | $7.9 \pm 0.5$ | $7.9 \pm 0.7$ | $11.0 \pm 1.2$ | $8.0 \pm 0.1$ | $15.6 \pm 0.7$ |
| ChannelViT-S ✳ | $25.0 \pm 2.8$ | $7.1 \pm 1.2$ | $8.2 \pm 1.1$ | $9.0 \pm 1.9$ | $8.0 \pm 0.1$ | $11.4 \pm 1.0$ |
| ChannelViT-S ♻ | $43.1 \pm 1.4$ | $7.9 \pm 0.5$ | $7.9 \pm 0.7$ | $11.0 \pm 1.2$ | $8.1 \pm 0.0$ | $15.6 \pm 0.7$ |
| ChannelViT-S ♻✳ | $25.0 \pm 2.8$ | $7.1 \pm 1.2$ | $8.2 \pm 1.1$ | $9.0 \pm 1.9$ | $8.0 \pm 0.1$ | $11.5 \pm 1.0$ |
| iBOT-MillionAID | $29.8 \pm 3.3$ | $5.3 \pm 0.5$ | $4.4 \pm 2.1$ | $10.0 \pm 1.9$ | $8.5 \pm 0.4$ | $11.7 \pm 0.95$ |
| iBOT-MillionAID ✳ | $10.3 \pm 1.6$ | $2.6 \pm 3.3$ | $2.7 \pm 3.2$ | $6.0 \pm 2.4$ | $7.8 \pm 0.4$ | $5.1 \pm 1.7$ |

Table 4: Description of Sentinel-1 and Sentinel-2 bands.

| Band name | Blue | Green | Red | Red Edge 1 |
|---|---|---|---|---|
| Name in Sentinel | B2 | B3 | B4 | B5 |
| Codename used in this paper | B | G | R | E1 |
| Band name | Red Edge 2 | Red Edge 3 | Near Infrared | Narrow Near Infrared |
| Name in Sentinel | B6 | B7 | B8 | B8a |
| Codename used in this paper | E2 | E3 | N | N' |
| Band name | Shortwave Infrared 1 | Shortwave Infrared 2 | C-Band VV | C-Band VH |
| Name in Sentinel | B11 | B12 | VV | VH |
| Codename used in this paper | S1 | S2 | VV | VH |

Table 5: Benchmark Results for Change Detection (LEVIR-CD, CDD) and Classification (RESISC45, UC Merced) tasks with Different Scale Distortions.

| **LEVIR-CD** | **1:1** | **1:2** | **1:4** | **1:8** | **AUC-F1** |
|---|---|---|---|---|---|
| iBOT-ImageNet | $90.7 \pm 0.1$ | $87.6 \pm 0.5$ | $40.2 \pm 12.0$ | $2.0 \pm 1.4$ | $63.3 \pm 2.5$ |
| iBOT-MillionAID | $90.6 \pm 0.2$ | $87.6 \pm 0.9$ | $50.4 \pm 15.1$ | $2.0 \pm 1.0$ | $65.2 \pm 3.2$ |
| SatlasPretrain (S2_SwinB_SI_RGB) | $87.1 \pm 3.2$ | $84.4 \pm 3.5$ | $51.5 \pm 12.4$ | $12.6 \pm 1.8$ | $64.6 \pm 2.9$ |
| GFM | $90.3 \pm 1.1$ | $88.6 \pm 1.0$ | $72.3 \pm 1.5$ | $6.2 \pm 1.1$ | $70.1 \pm 0.5$ |
| Prithvi | $85.2 \pm 0.1$ | $84.4 \pm 0.1$ | $76.4 \pm 1.1$ | $14.5 \pm 1.2$ | $69.1 \pm 0.4$ |
| **CDD** | | | | | **AUC-F1** |
| iBOT-ImageNet | $97.3 \pm 0.0$ | $96.6 \pm 0.0$ | $89.7 \pm 0.2$ | $76.9 \pm 0.4$ | $87.0 \pm 0.0$ |
| iBOT-MillionAID | $97.4 \pm 0.0$ | $96.8 \pm 0.0$ | $91.4 \pm 0.6$ | $79.2 \pm 0.9$ | $87.7 \pm 0.2$ |
| SatlasPretrain (S2_SwinB_SI_RGB) | $96.0 \pm 0.0$ | $95.1 \pm 0.0$ | $90.4 \pm 0.3$ | $82.7 \pm 0.4$ | $86.9 \pm 0.1$ |
| GFM | $96.8 \pm 0.0$ | $96.0 \pm 0.1$ | $88.9 \pm 0.3$ | $78.0 \pm 0.6$ | $86.6 \pm 0.2$ |
| Prithvi | $90.9 \pm 0.2$ | $90.5 \pm 0.2$ | $88.5 \pm 0.3$ | $82.9 \pm 0.8$ | $83.6 \pm 0.3$ |
| **RESISC45: full fine-tuning** | | | | | **AUC-ACC** |
| iBOT-ImageNet | $93.8 \pm 0.2$ | $84.9 \pm 0.8$ | $46.8 \pm 3.3$ | $18.1 \pm 0.7$ | $66.3 \pm 0.9$ |
| iBOT-MillionAID | $93.4 \pm 0.2$ | $84.3 \pm 1.2$ | $47.4 \pm 5.6$ | $18.7 \pm 2.0$ | $66.2 \pm 1.8$ |
| DINOv2 | $94.1 \pm 0.4$ | $84.3 \pm 1.7$ | $46.7 \pm 5.2$ | $19.3 \pm 2.6$ | $66.3 \pm 1.6$ |
| SatlasPretrain (S2_SwinB_SI_RGB) | $96.1 \pm 0.1$ | $89.2 \pm 1.2$ | $61.4 \pm 3.3$ | $23.7 \pm 2.6$ | $71.9 \pm 1.4$ |
| SatlasPretrain (Aerial_SwinB_SI) | $96.1 \pm 0.1$ | $89.2 \pm 0.6$ | $52.1 \pm 2.3$ | $14.9 \pm 1.5$ | $69.1 \pm 0.7$ |
| GFM | $95.7 \pm 0.1$ | $87.1 \pm 0.9$ | $57.4 \pm 3.4$ | $19.1 \pm 3.0$ | $69.7 \pm 1.0$ |
| **RESISC45: linear probing** | | | | | **AUC-ACC** |
| iBOT-ImageNet | $91.7 \pm 0.1$ | $89.3 \pm 0.2$ | $74.3 \pm 0.6$ | $40.2 \pm 0.9$ | $75.4 \pm 0.2$ |
| iBOT-MillionAID | $94.6 \pm 0.1$ | $92.2 \pm 0.2$ | $66.5 \pm 1.5$ | $25.1 \pm 1.3$ | $73.8 \pm 0.5$ |
| DINOv2 | $91.1 \pm 0.7$ | $87.2 \pm 1.0$ | $72.9 \pm 1.4$ | $40.3 \pm 1.0$ | $74.2 \pm 0.9$ |
| SatlasPretrain (S2_SwinB_SI_RGB) | $72.8 \pm 0.1$ | $58.0 \pm 0.2$ | $25.4 \pm 0.4$ | $15.0 \pm 0.3$ | $46.6 \pm 0.1$ |
| SatlasPretrain (Aerial_SwinB_SI) | $81.7 \pm 0.1$ | $65.7 \pm 0.1$ | $31.1 \pm 0.3$ | $15.1 \pm 0.1$ | $52.8 \pm 0.1$ |
| GFM | $91.1 \pm 0.0$ | $83.6 \pm 0.1$ | $64.9 \pm 0.4$ | $35.6 \pm 0.6$ | $70.8 \pm 0.2$ |
| **UC Merced: full fine-tuning** | | | | | **AUC-ACC** |
| iBOT-ImageNet | $98.6 \pm 0.7$ | $98.2 \pm 1.0$ | $91.0 \pm 2.7$ | $61.3 \pm 7.7$ | $86.2 \pm 1.9$ |
| iBOT-MillionAID | $98.7 \pm 0.8$ | $97.9 \pm 1.3$ | $84.3 \pm 4.3$ | $46.0 \pm 8.3$ | $82.9 \pm 1.0$ |
| DINOv2 | $98.1 \pm 0.5$ | $97.9 \pm 0.3$ | $98.1 \pm 0.4$ | $97.3 \pm 0.3$ | $91.8 \pm 0.1$ |
| SatlasPretrain (S2_SwinB_SI_RGB) | $98.7 \pm 0.2$ | $98.0 \pm 0.3$ | $87.3 \pm 2.6$ | $61.9 \pm 5.9$ | $85.5 \pm 1.3$ |
| SatlasPretrain (Aerial_SwinB_SI) | $99.1 \pm 0.2$ | $98.1 \pm 0.3$ | $86.1 \pm 3.1$ | $57.7 \pm 3.9$ | $84.9 \pm 0.9$ |
| GFM | $99.2 \pm 0.2$ | $98.3 \pm 0.6$ | $93.3 \pm 1.6$ | $69.9 \pm 3.8$ | $87.9 \pm 0.9$ |
| **UC Merced: linear probing** | | | | | **AUC-ACC** |
| iBOT-ImageNet | $98.0 \pm 0.3$ | $97.9 \pm 0.3$ | $91.8 \pm 0.7$ | $61.4 \pm 3.6$ | $86.1 \pm 0.5$ |
| iBOT-MillionAID | $99.5 \pm 0.1$ | $99.2 \pm 0.32$ | $75.7 \pm 2.9$ | $31.3 \pm 3.9$ | $80.2 \pm 0.7$ |
| DINOv2 | $97.4 \pm 0.2$ | $97.0 \pm 0.1$ | $96.8 \pm 0.1$ | $91.8 \pm 0.4$ | $90.3 \pm 0.1$ |
| SatlasPretrain (S2_SwinB_SI_RGB) | $85.7 \pm 0.8$ | $79.6 \pm 0.4$ | $55.6 \pm 1.6$ | $27.2 \pm 0.5$ | $65.1 \pm 0.3$ |
| SatlasPretrain (Aerial_SwinB_SI) | $95.0 \pm 0.3$ | $87.0 \pm 0.4$ | $67.0 \pm 0.8$ | $36.8 \pm 0.3$ | $73.5 \pm 0.3$ |
| GFM | $95.8 \pm 0.1$ | $93.9 \pm 0.2$ | $84.7 \pm 0.4$ | $47.7 \pm 0.4$ | $81.0 \pm 0.1$ |

Table 6: The impact of full fine-tuning on the loss of generalization capabilities. All models are iBOTs pretrained on MillionAID.

| **LEVIR-CD: full fine-tuning** | 1:1 | 1:2 | 1:4 | 1:8 | AUC-F1 |
|---|---|---|---|---|---|
| iBOT-MillionAID | $88.7 \pm 0.1$ | $86.5 \pm 0.2$ | $63.6 \pm 3.3$ | $7.5 \pm 0.5$ | $67.5 \pm 0.7$ |
| iBOT-MillionAID-augm | $90.6 \pm 0.2$ | $87.6 \pm 0.9$ | $50.4 \pm 15.1$ | $2.0 \pm 1.0$ | $65.2 \pm 3.2$ |
| **LEVIR-CD: frozen backbone** | | | | | |
| iBOT-MillionAID | $81.5 \pm 0.1$ | $81.0 \pm 0.4$ | $69.3 \pm 3.1$ | $17.0 \pm 7.9$ | $65.9 \pm 1.6$ |
| iBOT-MillionAID-augm | $84.4 \pm 0.0$ | $84.4 \pm 0.2$ | $61.6 \pm 7.8$ | $3.4 \pm 4.0$ | $64.7 \pm 2.0$ |
| **RESISC45: full fine-tuning** | | | | | AUC-ACC |
| iBOT-MillionAID | $94.6 \pm 0.2$ | $92.8 \pm 0.3$ | $70.4 \pm 4.0$ | $16.6 \pm 4.0$ | $73.7 \pm 1.3$ |
| iBOT-MillionAID-augm | $93.4 \pm 0.2$ | $84.3 \pm 1.2$ | $47.4 \pm 5.6$ | $18.7 \pm 2.0$ | $66.2 \pm 1.8$ |
| **RESISC45: linear probing** | | | | | |
| iBOT-MillionAID | $91.0 \pm 0.1$ | $87.5 \pm 0.1$ | $60.8 \pm 0.2$ | $9.3 \pm 0.2$ | $68.1 \pm 0.1$ |
| iBOT-MillionAID-augm | $94.6 \pm 0.1$ | $92.2 \pm 0.2$ | $66.5 \pm 1.5$ | $25.1 \pm 1.3$ | $73.8 \pm 0.5$ |
| **UC Merced: full fine-tuning** | | | | | |
| iBOT-MillionAID | $98.0 \pm 0.3$ | $97.2 \pm 0.6$ | $87.2 \pm 1.9$ | $38.7 \pm 3.0$ | $82.2 \pm 0.7$ |
| iBOT-MillionAID-augm | $98.7 \pm 0.8$ | $97.9 \pm 1.3$ | $84.3 \pm 4.3$ | $46.0 \pm 8.3$ | $82.9 \pm 1.0$ |
| **UC Merced: linear probing** | | | | | |
| iBOT-MillionAID | $96.9 \pm 0.0$ | $97.1 \pm 0.2$ | $93.6 \pm 0.2$ | $34.0 \pm 1.3$ | $82.5 \pm 0.2$ |
| iBOT-MillionAID-augm | $99.5 \pm 0.1$ | $99.2 \pm 0.32$ | $75.7 \pm 2.9$ | $31.3 \pm 3.9$ | $80.2 \pm 0.7$ |

Table 7: Dependence of the performance of fine-tuned models on sclae augmentation performed during pretraining and fine-tuning. All models are iBOTs trained on MillionAID.

| **Augmentation Phase** | 1:1 | 1:2 | 1:4 | 1:8 | |
|---|---|---|---|---|---|
| **LEVIR-CD** | | | | | AUC-F1 |
| Pretraining / Fine-tuning | $88.7 \pm 0.1$ | $86.5 \pm 0.2$ | $63.6 \pm 3.3$ | $7.5 \pm 0.5$ | $67.5 \pm 0.7$ |
| **Pretraining** / Fine-tuning | $90.6 \pm 0.2$ | $87.6 \pm 0.9$ | $50.4 \pm 15.1$ | $2.0 \pm 1.0$ | $65.2 \pm 3.2$ |
| Pretraining / **Fine-tuning** | $88.2 \pm 0.1$ | $88.4 \pm 0.1$ | $87.9 \pm 0.1$ | $86.1 \pm 0.1$ | $82.4 \pm 0.1$ |
| **Pretraining** / **Fine-tuning** | $89.9 \pm 0.1$ | $89.9 \pm 0.1$ | $89.4 \pm 0.1$ | $87.7 \pm 0.1$ | $83.9 \pm 0.1$ |
| **CDD** | | | | | AUC-F1 |
| Pretraining / Fine-tuning | $95.8 \pm 0.0$ | $95.3 \pm 0.0$ | $92.3 \pm 0.1$ | $80.1 \pm 0.5$ | $87.0 \pm 0.1$ |
| **Pretraining** / Fine-tuning | $97.4 \pm 0.0$ | $96.8 \pm 0.0$ | $91.4 \pm 0.6$ | $79.2 \pm 0.9$ | $87.7 \pm 0.2$ |
| **UC Merced** | | | | | AUC-ACC |
| Pretraining / Fine-tuning | $98.0 \pm 0.3$ | $97.2 \pm 0.6$ | $87.2 \pm 1.9$ | $38.7 \pm 3.0$ | $82.2 \pm 0.7$ |
| **Pretraining** / Fine-tuning | $98.7 \pm 0.8$ | $97.9 \pm 1.3$ | $84.3 \pm 4.3$ | $46.0 \pm 8.3$ | $82.9 \pm 1.0$ |
| Pretraining / **Fine-tuning** | $98.2 \pm 0.6$ | $98.3 \pm 0.6$ | $98.0 \pm 0.6$ | $95.7 \pm 1.2$ | $91.8 \pm 0.6$ |
| **Pretraining** / **Fine-tuning** | $95.3 \pm 1.8$ | $94.7 \pm 2.0$ | $94.0 \pm 2.4$ | $91.8 \pm 3.6$ | $88.4 \pm 2.1$ |

