# OpenReview forum: "Benchmarking Robustness of Foundation Models for Remote Sensing"
_ICLR.cc/2025/Conference — ICLR 2025 Conference Withdrawn Submission_

### Official Review · Reviewer_UcQC · 2024-10-15

**Soundness:** 2
**Presentation:** 3
**Contribution:** 1
**Rating:** 3
**Confidence:** 4

**Summary:**

The paper introduces a benchmark pipeline that measures the generalization ability and performances on downstream tasks. It first outlines the motivation for application of self-supervised foundation models to remotes sensing tasks and the needs to test these foundation models with different experimental settings. The paper points out a few significant axes of generalization and highlights why resolution and bands are of key interest. It explains the benchmarking methods and evaluation metrics of scale augmentation for resolution study and different spectral bands combination study. The authors select iBOT, a cutting-edge vision transformer variation, as backbone and tested the impact of introduction of scale augmentation at different training stages, they also develop a joint training strategy for mask decoder. The authors point out that fine-tuning has negative impact on the generalization performance, where frozen backbone can become handy. For the results, they list the metric scores for different models on different remote sensing tasks and demonstrate that most foundation models perform not very well on generalization to lower resolutions and different spectral bands, they also point out that froze backbone preserves general representations better than fine-tuned ones.

**Strengths:**

This paper gives out a comprehensive benchmarking for foundation models in aerial imagery processing. It addresses two practical challenges: resolution and spectral bands variation. It also evaluated different techniques which could cause positive/negative impact for generalization. It reveals the limitations of current models. This paper has a easy-to-understand writing style.

**Weaknesses:**

1. Lack of novelty: From what I received in this paper, this paper lacks new algorithm/architecture/methodology with fundamental contribution. All works are limited to testing and applying of architectures or metrics from previous works. Correct me if I am wrong.
2. New loss? New architecture?: This paper did a good job in demonstrating current models are not competitive in generalization to different resolutions and band combinations, but gives out no new solutions or possible proposals. For example, will it be helpful to develop a new loss mechanism or a new architecture? Or re-design current models to adapt to your generalization needs?
3. Multi/hyper-spectral data: Recent studies more focus on how to retrieve spectral representations across all bands instead of transferring the knowledge from, say RGB, to NIR. Correct me if I am wrong.
4. Pre-training, Fine-tuning: These are not new concepts at all and have been studied extensively, the conclusion of fine-tuned backbone impact generalization and frozen backbone preserves generalization ability is intuitive and familiar to most researchers and has been widely accepted as a fact, repeatedly conduct these trainings seems redundant.

**Questions:**

Benchmark is important for people to know what are the problems that causes poor generalization performances, but this paper seems to provide no useful information regarding why foundation models generalize bad?

All researchers know that large models suffer from overfitting and generalization issues, reiterate this with remote sensing models will not clarify it more. Instead, it would be more meaningful to explore what method could be used to address this issue? Did you or are you planning to propose any technique/algorithm/architecture design to address this generalization problem instead of simply freezing the pre-trained weights?

The claim that generalization to other geographical locations is hard to study seems a bit underwhelming since multiple datasets cover various landscapes under uniformed labeling framework such as EuroSAT/LoveDA? Also for seasonal generalization, as far as I know many satellite imagery sources such as Sentinel-2 provides data around the year. Correct me if I am wrong.

---

### Official Review · Reviewer_XFSf · 2024-10-28

**Soundness:** 2
**Presentation:** 3
**Contribution:** 2
**Rating:** 5
**Confidence:** 5

**Summary:**

The authors conduct benchmarking for remote sensing foundation models around different characteristics including changes to different resolutions and new bands.  The authors train several iBOT models on the MIllionAID dataset with different augmentations and evaluate it on change detection and scene classification.

**Strengths:**

This is an interesting and important line of investigation as the ability to produce high-quality remote sensing foundation models has dramatically lagged behind other domains.

Overall the text is clear and well wrtiten.

Significant benchmarks are run with multiple trials to generate error bars.

Design choices are all reasonable.

**Weaknesses:**

Only scene classification and change detection are explored.  There are many other tasks within remote sensing which are highly relevant and may have dramatically different feature relevancy (i.e. more spectrally dominant vs. structurally).

I disagree with the authors' stance that images at inference time are more likely to be lower-resolution.  Very often the opposite is true- there are huge amounts of publicly available low resolution imagery.  However, with the introduction of private satellite, planes, and drones, into the remote sensing space, people are trying to use these high-resolution sources for a very specific task while leveraging the decades of low resolution data that exists.  Similarly, satellite imagery is only going to get better over time so being able to adapt the low resolution data to a new high resolution task is paramount.

For tasks like change detection and scene classification in general, there is substantial labeled data out there- foundation models would hopefully boost performance, but they're still usable.  In contrast, there are many novel remote sensing tasks (especially in ecology/climate and agriculture) which are blocked by a lack of good labels and desperately need improved foundation models.  I think exploring some of these would dramatically improve the impact of the work.

Additional analysis and discussion is warranted around why contradictory results are seen for different tasks/datasets.  This type of result has been a key challenge (particularly in remote sensing), so I think it warrants more discussion.

**Questions:**

I don't have specific questions, but do feel like more discussion is needed around interpretation of the results.  For example, "we hypothesize that models pretrained in a self-supervised manner require less data..."- explain and discuss further.

---

### Official Review · Reviewer_DyRL · 2024-10-31

**Soundness:** 1
**Presentation:** 2
**Contribution:** 1
**Rating:** 1
**Confidence:** 4

**Summary:**

This paper introduces a benchmark aimed at evaluating the robustness and generalization capabilities of foundation models specifically designed for remote sensing applications. The authors assess performance across various image resolutions and spectral bands, focusing on tasks like change detection and scene classification. Using aerial imagery datasets such as RESISC45, BigEarthNet, LEVIR-CD, and OSCD, the paper benchmarks several models and explores the influence of pretraining and fine-tuning strategies on model robustness.

**Strengths:**

1. The paper's attempt to establish a generalization benchmark tailored to remote sensing fills an essential gap, addressing unique requirements in aerial imaging, like resilience to scale and spectral variations.
2. By focusing on change detection and scene classification, the paper makes an effort to connect benchmark results with real-world implications, enhancing the relevance of the benchmark.

**Weaknesses:**

1. Although the paper aims to establish a benchmark for evaluating foundation models in remote sensing, the scope of experimentation is limited. The study only explores a small subset of possible variations and factors within the remote sensing domain, which weakens its potential to serve as a comprehensive benchmark for ICLR readers. For instance, the study could have broadened its scope by including more downstream tasks, such as object detection and segmentation. Additionally, it fails to address important variables such as geographical location, times of the day, or seasonal variations, all of which could significantly impact the models' performance in real-world applications. By focusing on just a few experimental parameters, the study provides only a partial view of the potential applications and robustness of these foundation models in remote sensing.
2. Remote sensing data includes numerous factors that could influence model performance, such as resolution, spectral band variability, and environmental conditions. However, the current study only examines resolution and spectral bands, omitting several critical factors that users in the field of remote sensing would consider valuable. For instance, controlling for environmental factors (e.g., time of year, weather conditions) or ablation studies on model and dataset size could provide more meaningful insights. Additionally, benchmarks could have varied the type of pretraining datasets used, comparing options like FAIR1M and MillionAID, which cater specifically to different data characteristics. This limited approach leaves significant gaps in understanding the models' broader adaptability and robustness.
3. One of the most significant weaknesses of the paper is its lack of coverage of remote-sensing-specific foundation models, particularly those developed with unique properties for remote sensing challenges. Models such as Scale-MAE and SatMAE are designed to address the exact generalization challenges discussed in the paper, such as resilience to spatial, temporal, and spectral variability. Scale-MAE, for instance, is specifically designed to handle resolution variations, a critical factor in satellite imagery, while SatMAE incorporates temporal and locational encoding, making it suitable for applications that require adaptability across time and space. By not including these specialized models, the paper misses a critical opportunity to benchmark the very architectures that are purpose-built to handle the complexities of remote sensing, thus limiting the practical relevance and depth of the benchmark.
4. While the paper discusses self-distillation-based models, it does not sufficiently evaluate how different pretraining methods, such as DINOv2 and EVA, might impact model robustness and generalization capabilities. Advanced pretraining techniques are known to affect the performance of foundation models differently, especially when fine-tuning for specific tasks like those in remote sensing. Including a comparative analysis of these methods could offer insights into how various approaches to pretraining influence downstream task performance and generalization in remote sensing contexts. This omission reduces the value of the benchmark, as it does not provide a complete picture of how alternative pretraining methods might improve or hinder model performance in real-world scenarios.

### References
- [SatMAE] Cong, Yezhen, et al. "Satmae: Pre-training transformers for temporal and multi-spectral satellite imagery." Advances in Neural Information Processing Systems 35 (2022): 197-211.
- [Scale-MAE] Reed, Colorado J., et al. "Scale-mae: A scale-aware masked autoencoder for multiscale geospatial representation learning." Proceedings of the IEEE/CVF International Conference on Computer Vision. 2023.
- [EVA] Fang, Yuxin, et al. "Eva: Exploring the limits of masked visual representation learning at scale." Proceedings of the IEEE/CVF Conference on Computer Vision and Pattern Recognition. 2023.
- [DINOv2] Oquab, Maxime, et al. "Dinov2: Learning robust visual features without supervision." TMLR. 2024.

**Questions:**

Nothing in particular

---

### Official Review · Reviewer_t4Up · 2024-11-03

**Soundness:** 2
**Presentation:** 3
**Contribution:** 3
**Rating:** 3
**Confidence:** 5

**Summary:**

This paper examines the key characteristics and performance benchmarks for foundation models applied to remote sensing data. The authors present a comprehensive benchmark framework to assess the performance and robustness of these models across diverse scales and spectral bands. This benchmark includes tasks such as change detection and scene classification, leveraging publicly available datasets like RESISC45, BigEarthNet, LEVIR-CD, and OSCD. The topic is highly relevant, as establishing benchmarks for remote sensing (RS) foundation models is essential for advancing this field.

**Strengths:**

1) The topic of this paper is valuable, as benchmarking remote sensing (RS) foundation models is crucial for the future development of this field.
2) The experiments on spectral bands and the idea of a pretrained mask decoder are interesting, and I encourage the authors to expand and reorganize these sections. However, in its current form, I believe the paper still falls short of the standards required for ICLR.

**Weaknesses:**

1) I am concerned about the label quality of the datasets used. For example, BigEarthNet V2 [8] provides a significant improvement over BigEarthNet. While I understand that it might not be feasible to use BENv2 given its release date, the authors should consider data and label quality when selecting datasets for benchmarking. The authors shall discuss how they assessed the quality of the datasets used and what impact potential label quality issues might have on their results.

2) How do the authors define the generalizability of foundation models? Given that different foundation models are trained on different datasets or combinations, comparisons may be problematic. For instance, DinoV2 uses data augmentation techniques like random cropping and scaling, which enrich the spatial resolution of the training data. However, not all foundation models use such augmentations, potentially making direct comparisons unfair. The authors should address these inconsistencies and avoid drawing broad conclusions about generalizability without accounting for differences in training data and augmentations. The authors are suggested to provide a clear definition of generalizability in the context of their study and include a detailed analysis of the training data and augmentation techniques used by each evaluated model, and discuss how these differences might impact the comparisons and conclusions drawn from the benchmark results

3) It would be beneficial to also compare the foundation models with smaller models like ResNet50. This comparison could illustrate the advantages and trade-offs of using larger foundation models versus smaller ones.

4) The paper claims that the performance gap between frozen models and full fine-tuning is relatively large for models such as ChannelViT-S, Prithvi, and Clay v1. However, I am concerned about how the learning rate was selected. Using the same learning rate across different models may not be optimal. Moreover, different learning rates for the backbone and the task-specific heads (e.g., classification, segmentation, change detection) should be considered. Without optimizing these hyperparameters for each model, the conclusions drawn regarding their performance are not convincing.

5) The authors conclude that all tested models struggle with generalizability across scales and spectral bands. While foundation models are expected to generalize well to different data sources, this relies on training with large, diverse datasets. It is therefore expected that models trained on aerial images may perform poorly on satellite images due to the inherent differences between these data sources. I recommend that the authors further investigate the relationship between pre-training datasets, data augmentations, and model generalizability.

**Questions:**

1) The paper does not clearly justify the choice of datasets used for evaluation. Given that there are so many benchmark datasets available for remote sensing, why were these particular datasets chosen? A more thorough rationale is needed. Please elaborate on the specific criteria you have used for dataset selection. Please also discuss the strengths and limitations of the chosen datasets compared to alternatives, and how these choices might impact the generalizability of your results.

2) There are numerous well-known foundation models for remote sensing data, including vision-language models like RemoteCLIP [1] and Skyscript [2], multimodal models like DOFA [3], SSL4EO-S12 [4], and MMEarth [5], as well as others like msGFM [6] and SatMAE++ [7]. Considering the broad definition of a foundation model, why did the authors not include these in the evaluation? Please kindly explain the criteria for model selection and discuss how the inclusion or exclusion of specific models might affect your findings. Additionally, please kindly address the potential limitations of their current model selection in the paper's discussion section.

3) The evaluation tasks are limited to scene classification and change detection. Why were other critical tasks, such as semantic segmentation, object detection, and regression, not included? Please kindly justify your task selection and discuss how the inclusion of additional tasks like semantic segmentation or object detection might provide a more comprehensive evaluation of the foundation models' capabilities. Please kindly address this limitation in your paper and propose how future work could expand the range of tasks.

[1] Liu, Fan, et al. "Remoteclip: A vision language foundation model for remote sensing." IEEE Transactions on Geoscience and Remote Sensing (2024).

[2] Wang, Zhecheng, et al. "Skyscript: A large and semantically diverse vision-language dataset for remote sensing." Proceedings of the AAAI Conference on Artificial Intelligence. Vol. 38. No. 6. 2024.

[3] Xiong, Zhitong, et al. "Neural plasticity-inspired foundation model for observing the earth crossing modalities." arXiv preprint arXiv:2403.15356 (2024).

[4] Wang, Yi, et al. "SSL4EO-S12: A large-scale multimodal, multitemporal dataset for self-supervised learning in Earth observation [Software and Data Sets]." IEEE Geoscience and Remote Sensing Magazine 11.3 (2023): 98-106.

[5] Nedungadi, Vishal, et al. "MMEarth: Exploring multi-modal pretext tasks for geospatial representation learning." arXiv preprint arXiv:2405.02771 (2024).

[6] Han, Boran, et al. "Bridging remote sensors with multisensor geospatial foundation models." Proceedings of the IEEE/CVF Conference on Computer Vision and Pattern Recognition. 2024.

[7] Noman, Mubashir, et al. "Rethinking transformers pre-training for multi-spectral satellite imagery." Proceedings of the IEEE/CVF Conference on Computer Vision and Pattern Recognition. 2024.

[8] Clasen, Kai Norman, et al. "reben: Refined bigearthnet dataset for remote sensing image analysis." arXiv preprint arXiv:2407.03653 (2024).

---

### Note · Authors · 2024-12-04

**Comment:**

The concerns raised by reviewers raise a few categories of concerns: the scale of the tasks and datasets; the range of models tested; lack of algorithmic novelties (new losses, architectures etc.)

While we work on improving the tasks and datasets and try to increase the coverage of tested models (as much as our compute resources allow), we cannot understand the need of algorithmic novelties in a benchmark paper. We decided to withdraw the paper, but **it would be very beneficial for our future work if the reviewers and AC help us understand what is a scope of a good benchmark paper in ICLR**. ICLR's *Call for papers* includes a line "Datasets and Benchmarks". Should the papers in that category / topic necessarily include algorithmic innovations to be considered worthy for ICLR?

What are some benchmark papers published at previous ICLR conferences that the reviewers and ACs consider high quality?

We thank the reviewers for the comments and suggestions. We will use them in the next iterations of this work.

**Withdrawal Confirmation:**

I have read and agree with the venue's withdrawal policy on behalf of myself and my co-authors.